# A machine learning framework for the evaluation of myocardial rotation in patients with noncompaction cardiomyopathy

**Marcelo Dantas Tavares de Melo**[1], **Jose de Arimatéia Batista Araujo-Filho**[2], **José Raimundo Barbosa**[3], **Camila Rocon**[1,2], **Carlos Danilo Miranda Regis**[3], **Alex dos Santos Felix**[4], **Roberto Kalil Filho**[1,2], **Edimar Alcides Bocchi**[1], **Ludhmila Abrahão Hajjar**[1], **Mahdi Tabassian**[5], **Jan D'hooge**[5], **Vera Maria Cury Salemi**[1,2]*

**1** Heart Institute (InCor) do Hospital das Clínicas da Faculdade de Medicina da Universidade de São Paulo, São Paulo, Brazil, **2** Sírio Libanês Hospital, São Paulo, Brazil, **3** Federal Institute of Paraíba, João Pessoa, Brazil, **4** National Institute of Cardiology, Rio de Janeiro, Brazil, **5** Department of Cardiovascular Sciences, University of Leuven, Leuven, Belgium

* verasalemi@uol.com.br

## Abstract

### Aims

Noncompaction cardiomyopathy (NCC) is considered a genetic cardiomyopathy with unknown pathophysiological mechanisms. We propose to evaluate echocardiographic predictors for rigid body rotation (RBR) in NCC using a machine learning (ML) based model.

### Methods and results

Forty-nine outpatients with NCC diagnosis by echocardiography and magnetic resonance imaging (21 men, 42.8±14.8 years) were included. A comprehensive echocardiogram was performed. The layer-specific strain was analyzed from the apical two-, three, four-chamber views, short axis, and focused right ventricle views using 2D echocardiography (2DE) software. RBR was present in 44.9% of patients, and this group presented increased LV mass indexed (118±43.4 vs. 94.1±27.1g/m$^2$, $P = 0.034$), LV end-diastolic and end-systolic volumes ($P < 0.001$), E/e' (12.2±8.68 vs. 7.69±3.13, $P = 0.034$), and decreased LV ejection fraction (40.7±8.71 vs. 58.9±8.76%, $P < 0.001$) when compared to patients without RBR. Also, patients with RBR presented a significant decrease of global longitudinal, radial, and circumferential strain. When ML model based on a random forest algorithm and a neural network model was applied, it found that twist, NC/C, torsion, LV ejection fraction, and diastolic dysfunction are the strongest predictors to RBR with accuracy, sensitivity, specificity, area under the curve of 0.93, 0.99, 0.80, and 0.88, respectively.

### Conclusion

In this study, a random forest algorithm was capable of selecting the best echocardiographic predictors to RBR pattern in NCC patients, which was consistent with worse systolic,

**Data Availability Statement:** All relevant data are within the manuscript.

**Funding:** VMCS is supported by Conselho Nacional de Desenvolvimento Científico e Tecnológico

(CNPq) – grant number 307227/2018-9 The funders had no role in study design, data collection and analysis, decision to publish, or preparation of the manuscript.

**Competing interests:** The authors received no specific funding for this work and the authors have declared that no competing interests exist.

diastolic, and myocardium deformation indices. Prospective studies are warranted to evaluate the role of this tool for NCC risk stratification.

## Introduction

Noncompaction cardiomyopathy (NCC) is a genetic cardiomyopathy characterized by prominent left ventricular (LV) trabeculations, deep intertrabecular recesses communicating with the ventricular cavity, and a thin compacted external myocardial layer. Currently, there is no gold standard for the NCC diagnosis, and much debate and discussion persist regarding its classification, pathophysiology, and treatment of this entity [1].

The first echocardiographic description of NCC was reported in 1984, and the first echocardiographic diagnostic criterion was established in 1990 [2]. However, some healthy individuals can fulfill one or more echocardiographic criteria of NCC, and it is not clear if NCC is a distinct cardiomyopathy or an epiphenomenon of other cardiomyopathies [3]. The differential diagnosis with other cardiomyopathies is frequently challenging, and multimodality imaging correlations are usually required, leading to higher costs and extending time until the correct diagnosis is achieved. Although not considered a diagnostic criteria, van Dalen *et al.* observed that 88% of the NCC patients showed a loss of myocardial twist deformation of LV, with the rotation at the basal and apical levels predominantly in the same direction, a phenomenon called rigid body rotation (RBR) [4]. Another study by Peter *et al.* also detected this pattern in 53.3% of NCC patients [5], but the clinical relevance of these findings is still uncertain.

In the current era of precision medicine, machine learning techniques has been related to many potential applications in cardiac imaging [6]. Nevertheless, there are few recent papers that explored artificial intelligence tools in NCC patients, above all focused on refining the current diagnostic criteria [7] or on the prediction of adverse clinical outcomes [8] using data extracted from echocardiography and cardiac magnetic resonance, without correlations with myocardial strain analysis. Moreover, circumferential strain analysis software is frequently unavailable in most of echocardiographic machines and this evaluation must be performed offline in dedicated workstations, hampering its availability in small centers or beyond research field.

Although the role of myocardial strain analysis in risk stratification of patients with LV hypertrabeculation remain unknow [9], we hypothesize that RBR may have a potential value in the risk stratification of patients who fulfilled the clinical and imaging criteria of NCC. In that context, we developed and tested ML framework, consisting of a random forest algorithm and a neural network, to evaluate the best echocardiographic predictors to RBR pattern in NCC patients, providing additional information for risk stratification of these patients, without the necessity of image post processing in dedicated workstations.

## Materials and methods

This cross-sectional observational study was performed in a quaternary cardiology center where 50 outpatients with a high clinical pre-test probability diagnosis of NCC from 2016 to 2017 were studied [10]. All clinical and echocardiographic data were prospectively collected. NCC was diagnosed when patients fulfill all echocardiographic [11–13] and Petersen [14] criteria by cardiac magnetic resonance imaging, reinforcing the accuracy of this diagnosis in this sample.

Exclusion criteria included pregnancy, valvular heart disease (at least moderate), congenital heart disease, other associated cardiomyopathies, known coronary heart disease, pacemaker, and atrial fibrillation. According to Framingham score or angina, patients over age 40, if they had a moderate risk of coronary artery disease, underwent a non-invasive or invasive coronary artery study. All patients had a complete neurological examination, and all first relatives' members of patients were also recruited for NCC screening.

Data were expressed as mean ± SD and frequency (percentage). Comparisons between patients with and without RBR were performed using 2-sample *t*-test (or Wilcoxon rank-sum test) and χ2 tests (or Fisher exact test) for continuous and categorical data, respectively. All statistical analyses were performed using R version 4.0.0 (R Foundation for Statistical Computing, Vienna, Austria), and a *P*-value of < 0.05 was considered statistically significant. To calculate the correlation matrix, the Pearson form was used, in which the correlation coefficient **r** for each pair of features *x* and *y* are calculated.

The institutional review board (Comissão de Ética para Análise de Projeto de Pesquisa–CAPPesq, number 0103/09) approved this study for human subject studies, and all participants provided written informed consent before enrollment.

## Echocardiographic imaging protocol

Comprehensive echo studies were performed with Vivid E9 echocardiographic ultrasound system (GE Healthcare, Norway). The exams were analyzed by a single trained sonographer, following the European Association of Echocardiography/American Society of Echocardiography guidelines for cardiac chambers and functional analysis [15]. Jenni criterion was used to calculate the N/C ratio, which measures the maximal end-systolic ratio of non-compacted to compacted layers [12]. Data were exported to a dedicated workstation (EchoPac 202, GE Vingmed, Horten, Norway) for off-line analysis by a blinded observer.

## Bidimensional left ventricular layer-specific strain

Quantification of layer-specific strain measurements was performed offline with dedicated software (EchoPAC V.202, GE). For speckle tracking echocardiography (STE) and longitudinal strain analysis, digital loops of the right ventricle were obtained from apical 4-chamber and/or right ventricle-focused apical 4-chamber views, and left ventricle (LV) from apical 3-chamber, 2-chamber, and 4-chamber views. For radial and circumferential LV strain, parasternal short axis acquisitions were obtained from the mitral valve, papillary muscle, and apical levels, using mitral valve and apical values to obtain twist and torsion. Three cardiac cycles were acquired from each view at a frame rate of 40–80 frames/sec in patients in sinus rhythm.

Preliminary longitudinal strain analysis was performed online in the ultrasound machine, checking if image quality was good enough to permit adequate tracking of myocardial acoustic markers (speckles) during the entire cardiac cycle. STE analysis was performed semi-automatically, after the operator's manual setting of 3 points on the endocardial border (2 basal and 1 at the apex). When the region of interest (ROI) included the whole thickness of the ventricle and excluded other structures (such as trabeculae, moderator band, and valvular tissue), the processing was started, and analysis proceeded on a frame-to-frame basis using an automatic tracking system. The ROI generated by the software included basal, mid, and apical segments of opposed walls, divided in 6 segments. Longitudinal peak strain values were measured for each segment, and global longitudinal strain values were calculated by averaging the values. The investigators were blinded to these initial results until the offline analysis of the remaining parameters was performed. Radial and circumferential strains were analyzed exclusively offline. For radial and circumferential strain analysis, the endocardial border was traced just

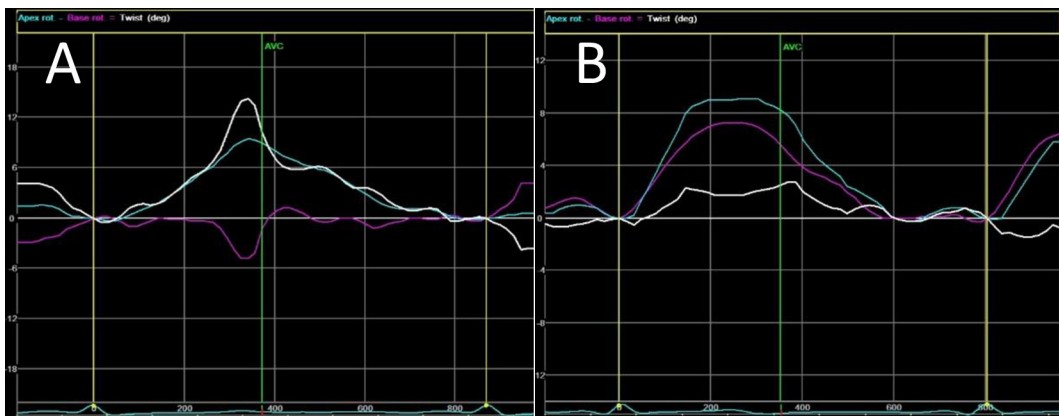

**Fig 1. Apical rotation and basal rotation curve obtained from NCC patients.** Normal left ventricular torsion (A), and rigid body rotation pattern (B). The green line represents the apex rotation, and the basal rotation is showed in pink, while the curves mean represented in white. Ordinate axis with positive value expresses counterclockwise rotation and with a negative value, clockwise rotation. AVC, aortic valve closure; LV twist = apical rotation–basal rotation (white line).

within the endocardium using point-by-point tracing, and particular care was taken to adjust the tracking of all segments. A second larger concentric circle was then automatically generated and manually adjusted near the epicardium such that the area of interest included the entire myocardial wall. The image was then played so that tracking in the region of interest could be fine-tuned by visual assessment to ensure that all wall segments were tracked appropriately throughout the cardiac cycle and that the sectors defining each wall segment were adjusted properly.

Global radial and circumferential strains were measured as the average of the 6 regional segments from the parasternal short-axis view at the mitral valve level, papillary muscles, and LV apex. Left ventricular twist was calculated as the relative rotation of the apex around the LV long axis with respect to the base during the cardiac cycle, and torsion as the twist value normalized to the distance between LV apex and base, expressed in degrees per centimeter (˚/cm). LV torsion was considered normal whenever LV presented with initial counterclockwise basal and clockwise apical rotation, followed by end-systolic clockwise basal and counterclockwise apical rotation, and RBR when LV showed rotation at the basal and apical level predominantly in the same direction as demonstrated in Fig 1.

## Data processing

Fig 2 demonstrates the structure of the proposed ML framework and the main stages involved in its implementation. In this section, these stages will be explained in complete detail. We include the following echocardiogram parameters in this model: twist, torsion, left ventricular ejection fraction, left ventricular global longitudinal strain, left ventricular circumferential strain (apical, mid and basal), left ventricular global circumferential strain, left ventricular radial strain (apical, mid and basal), left ventricular global radial strain, TAPSE, e', left atrial strain (reservoir, conduit and buster), left ventricular eccentric and concentric hypertrophy, tricuspid velocity peak, isovolumic relaxation time, right ventricular free wall strain, right ventricular S', relative wall thickness, A wave velocity, E wave velocity, E/A ratio, E/e', diastolic dysfunction left ventricular length, deceleration time, fractional area change, left atrial volume index, right ventricle (base region), left ventricular diastolic and systolic diameters, left ventricular end systolic and end diastolic index volume, NC/C ratio, RBR pattern.

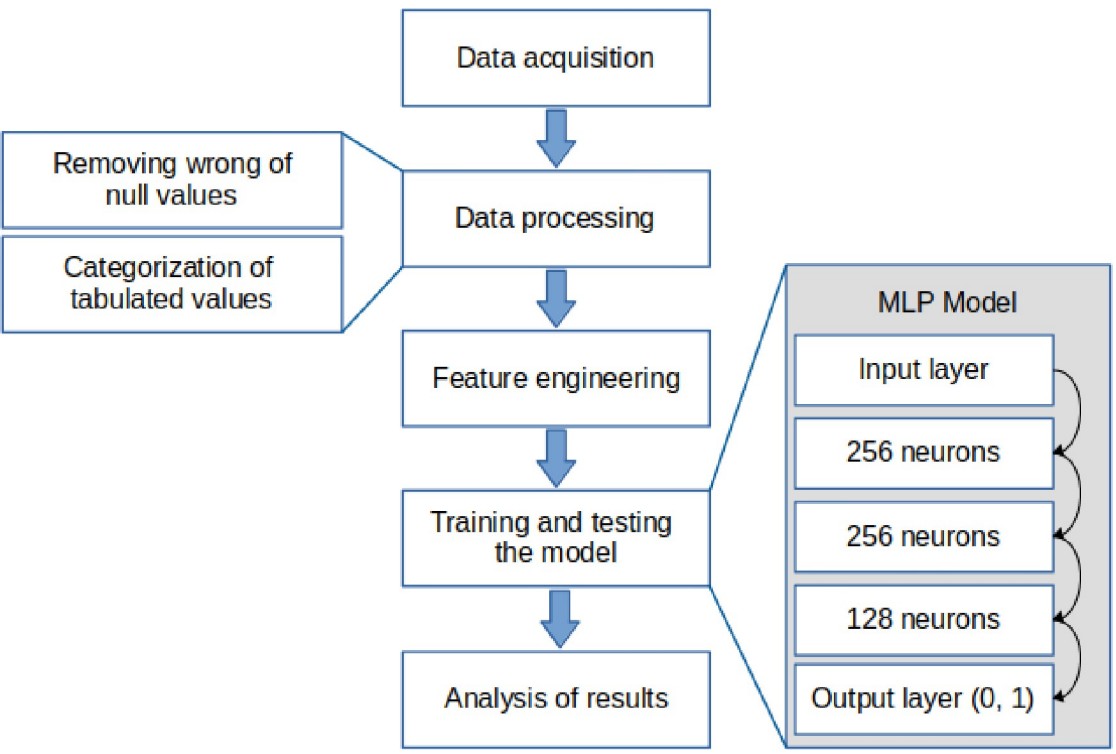

**Fig 2. Task execution diagram from the data acquisition to the results of noncompaction cardiomyopathy patients.**

Initially, the raw data samples were pre-processed to provide a set of input samples that can be used for developing an efficient learning model. For this purpose, features (columns) or samples (lines) that had many null values or values outside the range were eliminated from the bank. Both situations, removing columns or rows, were carefully analyzed because removing a critical column can harm the model's performance, whereas removing rows can reduce the size of the bank, making the training and testing stage unfeasible.

The number of features was also reduced by feature engineering techniques that allow the choice of the best features based on their influence on the performance of a learning model. A feature selection method based on a random forest algorithm was used [16]. This method indicates how useful a feature is for building the classifier. A tree was created for each feature, and then the number of optimization divisions for each tree is observed. The performance measure consists of the error function of the tree by the number of divisions.

## Multilayer perceptron neural networks

A neural network model called Multi-Layer Perceptron (MLP) was used. This type of neural network was selected to solve complicated classification problems when the amount of data for training the network is limited. However, the proposed ML framework can also be implemented using a state-of-the-art deep neural network when an extensive dataset for properly training such a network is available.

For adequate configuration of a neural network, layers, and neurons, it is necessary to consider factors such as the quality and quantity of samples from the dataset and the nature of the addressed problem. However, a close configuration is possible based on an empirical approach and observing criteria like regular distribution of layers and error rate. A small number of

hidden layers can generate an inaccurate model, whereas too many layers can cause overfitting. Also in Fig 2, the input layer contains the number of neurons corresponding to the features; the hidden layers are distributed in 3 layers, with the same activation function, rectified linear unit (ReLU); however, the output layer counts on 2 neurons and uses a sigmoidal activation function.

The Scikit-learn library was used to provide the implementation of this neural network. A k-fold validation approach with k = 5 was used to validate the model, corresponding to a training dataset with 60% of the samples, testing with 20%, and validation with 20%. The training stages of 3000 times each, and the learning rate is equal to 0.001.

## Results

Table 1 summarizes the demographic data in the present study. A total of 50 NCC patients were recruited, and one patient was excluded because more than 2 segments of a 17-segment model had low quality for strain analysis. Table 1 summarizes the demographic data in the entire sample (n = 49) and in the patients with (n = 22) or without (n = 27) LV RBR. Twenty-one men (42.8 ± 14.8 years) were included in the sample. The most common cardiovascular risk factor was systemic arterial hypertension (26.5%); most patients were in NYHA I (79.6%); family history of NCC was present in almost half of patients (49%). Anticoagulation therapy

**Table 1. Demographic, clinical, and laboratory data of noncompaction patients.**

| Patient characteristics | | All patients (*N* = 49) | LV-RBR absent (*n* = 27) | LV-RBR present (*n* = 22) | *P* -value |
|---|---|---|---|---|---|
| Gender (male), n (%) | | 21 (42.9) | 10 (37.0) | 11 (50.0) | 0.534 |
| Age (years) | | 42.8 (14.8) | 43.8 (12.6) | 41.5 (17.3) | 0.596 |
| Body surface area (m$^2$) | | 1.76 (0.21) | 1.81 (0.17) | 1.69 (0.24) | 0.068 |
| NYHA, n (%) | I | 39 (79.6) | 22 (81.5) | 17 (77.3) | 0.868 |
| | II | 7 (14.3) | 3 (11.1) | 4 (18.2) | 0.238 |
| | III | 3 (6.12) | 2 (7.41) | 1 (4.55) | 0.037 |
| Diabetes mellitus, n (%) | | 3 (6.12) | 3 (11.1) | 0 (0.00) | 0.242 |
| Smokers, n (%) | | 3 (6.12) | 2 (7.41) | 1 (4.55) | 1.000 |
| Systemic arterial hypertension, n (%) | | 13 (26.5) | 8 (29.6) | 5 (22.7) | 0.827 |
| Ventricular tachycardia, n (%) | | 7 (14.3) | 4 (14.8) | 3 (13.6) | 1.000 |
| Family history of NCC, n (%) | | 24 (49.0) | 15 (55.6) | 9 (40.9) | 0.464 |
| History of embolic events, n (%) | | 1 (2.04) | 0 (0.00) | 1 (4.55) | 0.449 |
| ACE-inhibitor, n (%) | | 42 (85.7) | 22 (81.5) | 20 (90.9) | 0.436 |
| **Anticoagulation, n (%)** | | **19 (38.8)** | **6 (22.2)** | **13 (59.1)** | **0.019** |
| Beta blocking agent, n(%) | | 39 (79.6) | 19 (70.4) | 20 (90.9) | 0.152 |
| Serum creatinine (mg/dL) | | 0.90 (0.20) | 0.91 (0.21) | 0.88 (0.19) | 0.593 |

ACE: angiotensin-converting enzyme; LV-RBR: left ventricular rigid body rotation; NCC: noncompaction cardiomyopathy; NYHA: New York Heart Association (functional class).

was statistically significant in the left ventricular rigid body rotation (LV-RBR) group (90.9% vs. 70.4%, $P$ = 0.019).

A total of 38 echocardiographic features were extracted from the dataset, as described in Table 2. Rigid body rotation pattern was present in 44.9% of patients. Left ventricular remodeling assessed by LV mass indexed (118±43.4 vs. 94.1±27.1g/m$^2$, $P$ = 0.034), left ventricular diastolic diameter (55.4±7.45 vs. 49.3±6.20mm, $P$ = 0.004) and the LV end-diastolic volume index (92.7±38.9 vs. 53.6±14.1mL/m$^2$, $P$ < 0.001) were higher in the RBR group. Also, E/e' was increased (12.2±8.68 vs. 7.69±3.13, $P$ = 0.034) and LV ejection fraction (40.7±8.71 vs. 58.9 ±8.76%, $P$ < 0.001) was lower in patients with RBR. Left ventricular mechanical parameters such as global longitudinal, radial, and circumferential strain were more affected in the RBR group, whereas noncompaction/compaction ratio (NC/C) presented no difference. On the other hand, LV diastolic parameter, left atrial strain, and right ventricle free wall strain (RV FWS) were not statistically different between the groups (Table 2).

The correlation matrix of the used dataset is shown in the Fig 3; it is possible to identify a set of features that have a high correlation with RBR. The strongest ones were torsion, twist ($r$ = -0.65), NC/C ($r$ = 0.62), and LVEF ($r$ = -0.51). Additional findings, the ratio NC/C had the following correlations: LVEF ($r$ = - 0.77), LV diastolic dysfunction ($r$ = 0.63), LV GLS ($r$ = -0.62), torsion ($r$ = -0.61), and twist ($r$ = -0.60).

Fig 4 presents the features in decreasing order, according to the influence on the Random Forest classification. The features marked as blue were selected based on the feature selection algorithm, and the feature marked as green was selected based on medical decision.

Ten features were selected based on the results obtained after applying the importance ranking and recursive elimination methods. One additional feature LV GLS was included in the model due to its clinical relevance. This type of feature plays an essential role during the classification. However, the algorithm can ignore this importance due to factors like small samples and slight variation in the value of the feature in the samples, justifying the correction by medical analysis in specific occasion. Finally, the features marked in red were removed from the model formation after the application of feature selection and medical analysis.

Aiming to select which parameters influence the RBR status in NCC patients, our model achieved high accuracy, sensitivity, and specificity: 0.93, 0.99, and 0.88 using the dataset with features selection approach. The area under the curve (AUC) in Fig 5 illustrates the performance of our model, which was 0.92.

The classification was also performed using the database with all the features. This execution aimed to identify whether the application of the features selection technique had positive results in the model's performance. The results obtained for accuracy, sensitivity, and specificity: 0.86, 0.88, and 0.83, respectively.

## Discussion

To the best of our knowledge, this is the first study assessing RBR by LV mechanics in NCC patients using a neural network to understand the intricate interaction between different echocardiographic parameters. For this, we have analyzed which echocardiographic parameters can consistently predict the presence of RBR, providing a new approach in the pathophysiological mechanism of LV contraction in NCC patients.

Left ventricular torsion occurs as a balance between the interaction of endocardial and epicardial fibers [17]. It is acceptable that the disappearance of torsion could increase endocardial stress and strain, increasing heart oxygen demand [18]. Conversely, NCC may present a distinct pattern [4], characterized by clockwise basal and apical rotation throughout systole. The excess of trabeculation affects the endocardial layer. Thereby, the more trabeculated is the

**Table 2. Echocardiographic parameters evaluated in noncompaction cardiomyopathy with rigid body rotation and included in the prediction model.**

| Echocardiographic features, n (%) | | All patients (N = 49) | LV-RBR absent (n = 27) | LV-RBR present (n = 22) | P-value |
|---|---|---|---|---|---|
| LV mass (g) | | 183 (63.3) | 171 (54.1) | 198 (71.6) | 0.152 |
| **LVIM (g/m²)** | | 105 (36.9) | 94.1 (27.1) | 118 (43.4) | 0.034 |
| **LVDD (mm)** | | 52.0 (7.39) | 49.3 (6.20) | 55.4 (7.45) | 0.004 |
| RV basal (mm) | | 34.9 (6.16) | 34.0 (4.83) | 36.0 (7.44) | 0.265 |
| **TAPSE (mm)** | | 21.7 (4.35) | 23.2 (3.53) | 19.7 (4.55) | 0.005 |
| S' RV (cm/s) | | 13.2 (2.64) | 13.8 (2.06) | 12.3 (3.10) | 0.068 |
| FAC (%) | | 49.2 (10.8) | 47.3 (8.56) | 51.4 (12.8) | 0.202 |
| LAI (mL/m²) | | 35.2 (14.8) | 32.6 (14.1) | 38.4 (15.4) | 0.184 |
| E (cm/s) | | 76.4 (23.9) | 73.1 (21.3) | 80.5 (26.8) | 0.314 |
| A (cm/s) | | 59.1 (19.7) | 57.5 (17.1) | 61.2 (22.8) | 0.532 |
| E/A | | 1.37 (0.44) | 1.32 (0.36) | 1.43 (0.53) | 0.416 |
| DT (ms) | | 219 (85.8) | 202 (78.3) | 239 (92.3) | 0.155 |
| TRIV (ms) | | 111 (25.3) | 109 (28.6) | 113 (21.0) | 0.640 |
| e' (cm/s) | | 9.78 (4.40) | 10.7 (4.43) | 8.60 (4.16) | 0.097 |
| **E/e'** | | 9.73 (6.58) | 7.69 (3.13) | 12.2 (8.68) | 0.031 |
| Tricuspid velocity peak (m/s) | | 3.26 (4.67) | 2.46 (0.68) | 4.02 (6.47) | 0.354 |
| **LV EDVI (mL/m²)** | | 71.1 (34.0) | 53.6 (14.1) | 92.7 (38.9) | <0.001 |
| **LV ESVI (mL/m²)** | | 38.1 (28.0) | 22.5 (9.71) | 57.2 (31.3) | <0.001 |
| **LVEF (%)** | | 50.7 (12.6) | 58.9 (8.76) | 40.7 (8.71) | <0.001 |
| **LV GLS (%)** | | 15.4 (4.67) | 17.9 (3.74) | 12.2 (3.72) | <0.001 |
| **LV Twist (°)** | | 9.40 (8.32) | 15.2 (6.25) | 2.32 (3.82) | <0.001 |
| **LV Torsion (°/cm)** | | 1.16 (1.05) | 1.89 (0.81) | 0.27 (0.45) | <0.001 |
| LAS reservoir (%) | | 30.6 (11.5) | 32.3 (10.8) | 28.5 (12.2) | 0.256 |
| LAS booster (%) | | 12.1 (6.09) | 12.6 (6.05) | 11.5 (6.22) | 0.517 |
| LAS conduit (%) | | 18.1 (9.40) | 19.6 (9.22) | 16.2 (9.50) | 0.213 |
| RV FWS (%) | | 23.0 (7.50) | 24.1 (5.55) | 21.6 (9.30) | 0.265 |
| **LVRS basal (%)** | | 26.3 (14.9) | 33.2 (14.0) | 17.9 (11.6) | <0.001 |
| LVRS mid (%) | | 26.9 (15.5) | 30.0 (15.4) | 23.1 (15.1) | 0.121 |
| LVRS apical (%) | | 22.8 (18.5) | 23.7 (13.7) | 21.7 (23.4) | 0.720 |
| **LV GRS (%)** | | 25.3 (11.5) | 29.0 (10.8) | 20.9 (11.1) | 0.014 |
| **LVCS basal (%)** | | 11.0 (6.32) | 13.2 (4.95) | 8.16 (6.79) | 0.006 |
| **LVCS mid (%)** | | 11.5 (6.18) | 13.2 (4.76) | 9.53 (7.18) | 0.049 |
| **LVCS apical (%)** | | 13.7 (7.69) | 17.0 (6.23) | 9.57 (7.44) | 0.001 |
| **LV GCS (%)** | | 12.1 (6.16) | 14.5 (4.37) | 9.22 (6.88) | 0.004 |
| **Dilated LV** | | 0.22 (0.42) | 0.11 (0.32) | 0.36 (0.49) | 0.046 |
| NC/C | | 2.99 (±0.99) | 2.3 (±0.15) | 3.72 (±0.98) | 0.13 |
| LV (%) remodeling | CH | 6 (12.2) | 4 (14.8) | 2 (9.09) | 0.940 |
| | CR | 4 (8.16) | 2 (7.41) | 2 (9.09) | 0.572 |
| | EH | 16 (32.7) | 8 (29.6) | 8 (36.4) | 0.940 |
| | normal | 23 (46.9) | 13 (48.1) | 10 (45.5) | 0.957 |

CH, concentric hypertrophy; CR, concentric remodeling; DT, deceleration time; EH, eccentric hypertrophy; FAC, fractional area change; IRVT, isovolumic relaxation time; LAI, left atrium volume indexed; LAS, left atrium strain; LV, left ventricle; LVDD, left ventricular diastolic diameter; LV EDVI, left ventricular end-diastolic volume indexed; LVEF, left ventricle ejection fraction; LVCS, left ventricle circumferential strain; LV ESVI, left ventricular end-systolic volume indexed; LV GCS, left ventricular global circumferential strain; LV GLS, left ventricular global longitudinal strain; LV GRS, left ventricular global radial strain; LVIM, left ventricular mass indexed; LVRS, left ventricular radial strain; NC/C, noncompacted/compacted ratio; RBR, left ventricular rigid body rotation; RV, right ventricle; RV FWS, right ventricle free wall strain; TAPSE, tricuspid annular plane systolic excursion.

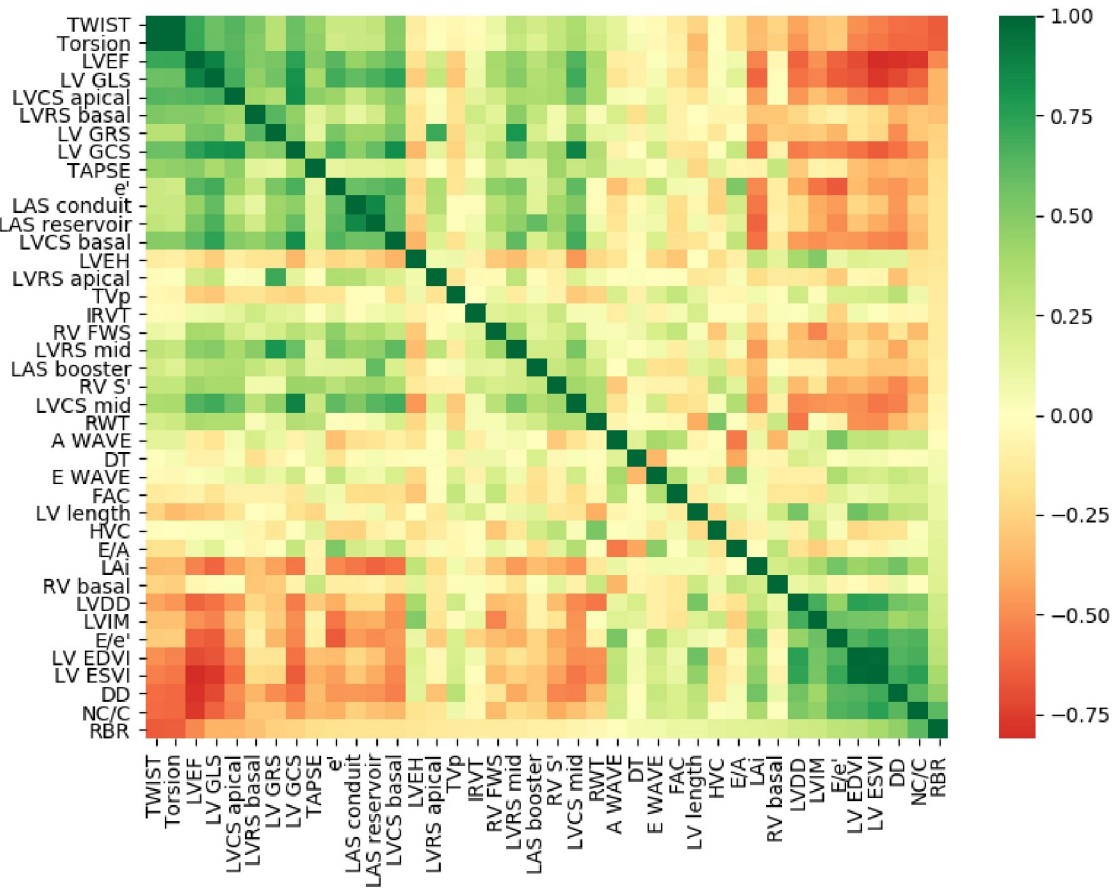

**Fig 3. Pearson's correlation matrix is ordered from the coefficients between the features and rigid body rotation.**

LV, the more severe endocardial fibers are affected. It could be an explanation for why RBR pattern is more prevalent in NCC patients.

Recently, Sabatino *et al.* proposed a discriminative value of LV twist in the NCC diagnosis [19]. Despite that, twist analysis has limited availability because it is laborious and not

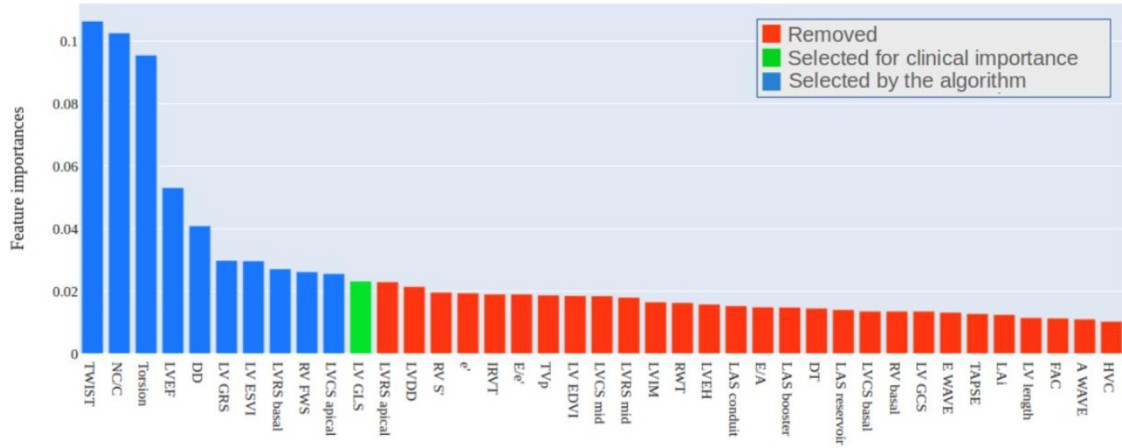

**Fig 4. Echocardiographic features in decreasing order of importance to rigid body rotation in noncompaction cardiomyopathy patients.**

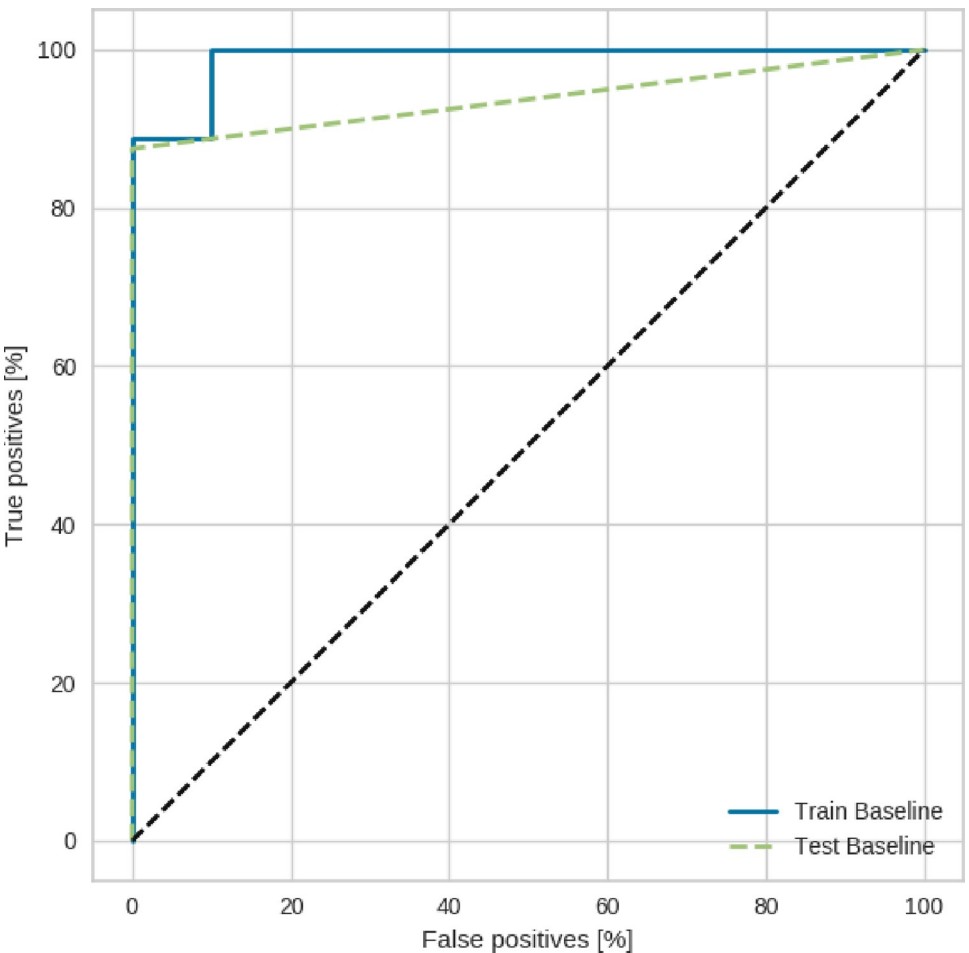

**Fig 5. ROC curves for prediction model to rigid body rotation pattern in noncompaction cardiomyopathy patients.** The continuous blue line shows the ROC curve obtained from the model's performance during the training stage. The dashed orange line shows the ROC curve obtained during the test step.

incorporated into most echocardiographic machines. Also, images acquisition/storage in a workstation offline is needed. Computational models could automatize these measurements and the algorithms are sufficiently close to automated measurements, reducing the time spent on analysis and errors related to manual calculations [20], especially in NCC patients.

Much debate persists regarding the clinical relevance of RBR pattern in NCC patients. Peters *et al.* showed that RBR was not associated with more adverse remodeling than in subjects with LVNC and normal LV rotation [5]. However, different from our study, they showed that LVEF was severely reduced in both groups (LVEF 27.9 ± 9.7 vs. 24.9 ± 11.7), making it more difficult to find consistent differences in LV remodeling. Of note, the RBR pattern was found in 53.3% of patients, remarkably close to our findings (44.9%). On the other hand, we found that RBR pattern was associated with worse LV remodeling and lower LVEF, increased ventricular volumes and mass, which is more consistent with previous studies in other cardiomyopathies [21, 22]. Interestingly, our matrix correlation and AI model recognized that RBR was correlated with 5 relevant echocardiographic parameters: LVEF, diastolic dysfunction twist, NC/C ratio and torsion.

Nowadays, which mechanical parameters have prognostic relevance in NCC patients are considered an open question. Intriguingly, the Multi-Ethnic Study of Atherosclerosis (MESA)

study, designed to investigate the prevalence and progression of subclinical atherosclerosis (not including NCC patients), found that LV function was worse in individuals with greater LV trabeculation [9]. Gastl *et al.* showed that cardiac magnetic resonance imaging derived deformation indices may show added value to assess functional impairment in LVNC, regardless of LVEF [23]. We found that the NC/C ratio correlates with RBR, LVEF, and diastolic dysfunction, which adds a coherent link between excess trabeculation and LV mechanical disturbance. Prospective studies are warranted to further explore these correlations in order to find new diagnostic and prognostic markers in NCC patients.

Our findings might suggest new insights into progressive myocardial dysfunction and the pathogenesis of NCC. We emphasize that these tools are not ready for clinical use, but we think they may impact the clinical decision-making in the near future. We also recognize that our results should be interpreted as an exploratory analysis, and they are not a substitute for the complete analysis of myocardial strain when available. Prospective studies with larger patient cohorts are needed to further validate the prognostic implication of our data, and it lends support for the role of this technique in refining the risk stratification of NCC patients.

## Limitations

This study has some limitations. First, a small sample size limits the application of more advanced neural networks. Second, we could not explore quantitatively the parameters that caused more influence in RBR, considering our small sample size. Third, all analysis was performed in only one software, we could not test the algorithm with different vendors which reduce its applicability. While there is a joint effort among the vendors to diminish these discrepancies [24]. The database used has a challenging configuration when working with machine learning techniques such as neural networks due to the number of samples and features [25]. Nevertheless, it was possible to use feature engineering techniques that make it possible to adjust the database for the desired application, maintaining the consistency of the results.

## Conclusion

Rigid body rotation was associated with pronounced LV remodeling and dysfunction. A machine learning model could identify the 11 parameters that consistently predict the presence of RBR in NCC patients. Further prospective studies should be addressed in order to investigate the role of RBR in the diagnosis and prognostication of NCC patients.

## Author Contributions

**Conceptualization:** Marcelo Dantas Tavares de Melo, Vera Maria Cury Salemi.

**Data curation:** Marcelo Dantas Tavares de Melo, José Raimundo Barbosa, Carlos Danilo Miranda Regis, Alex dos Santos Felix, Mahdi Tabassian.

**Formal analysis:** Marcelo Dantas Tavares de Melo, Jose de Arimatéia Batista Araujo-Filho, Mahdi Tabassian, Jan D'hooge.

**Investigation:** Vera Maria Cury Salemi.

**Methodology:** Marcelo Dantas Tavares de Melo.

**Project administration:** Vera Maria Cury Salemi.

**Resources:** Marcelo Dantas Tavares de Melo.

**Supervision:** Ludhmila Abrahão Hajjar, Vera Maria Cury Salemi.

**Validation:** Marcelo Dantas Tavares de Melo, José Raimundo Barbosa.

**Writing – original draft:** Marcelo Dantas Tavares de Melo.

**Writing – review & editing:** Jose de Arimatéia Batista Araujo-Filho, Camila Rocon, Carlos Danilo Miranda Regis, Alex dos Santos Felix, Roberto Kalil Filho, Edimar Alcides Bocchi, Ludhmila Abrahão Hajjar, Mahdi Tabassian, Jan D'hooge, Vera Maria Cury Salemi.

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
