## [Decision Letter · Decision Letter 0]

16 Jun 2021

PONE-D-21-10289

A Machine learning framework for the evaluation of myocardial dysfunction in patients with noncompaction cardiomyopathy and rigid body rotation

PLOS ONE

Dear Dr. Salemi,

Thank you for submitting your manuscript to PLOS ONE. After careful consideration, we feel that it has merit but does not fully meet PLOS ONE’s publication criteria as it currently stands. Therefore, we invite you to submit a revised version of the manuscript that addresses the points raised during the review process.

We look forward to receiving your revised manuscript.

Kind regards,

Zhifan Gao

Academic Editor

PLOS ONE

Journal Requirements:

2. Thank you for including your ethics statement: This study was approved by the institutional review board for human subject studies, and all participants provided written informed consent prior to enrollment.

 [No].

[No].

Additional Editor Comments (if provided):

Reviewers' comments:

Reviewer's Responses to Questions

**Comments to the Author**

1. Is the manuscript technically sound, and do the data support the conclusions?

Reviewer #1: Partly

Reviewer #2: Partly

2. Has the statistical analysis been performed appropriately and rigorously? 

Reviewer #1: Yes

Reviewer #2: Yes

3. Have the authors made all data underlying the findings in their manuscript fully available?

Reviewer #1: No

Reviewer #2: No

4. Is the manuscript presented in an intelligible fashion and written in standard English?

Reviewer #1: Yes

Reviewer #2: Yes

5. Review Comments to the Author

Reviewer #1: In this study, authors presented a study of NCC identification by using ML method. They focused on evaluation of the RBR criteria. They stated that good performance was obtained. However, there were several major issues that the result was therefore not convincing.

Comments:

1. The language usage throughout this paper need to be improved, the author should do some proofreading on it. Give the article a mild language revision to get rid of few complex sentences that hinder readability and eradicate typo erros.

2. Overall, the basic background is not introduced well. To my understanding, differentiation between NCC patient to non NCC patient was the key objective. However, no explanation about the ‘3 most common criteria’ was provided while limited information about the RBR without explanation about the correlation between RBR and NCC and the correlation between RBR and other echocardiographic parameters. It was confusing that whether the NCC or the RBR was the objective of this work.

3. Introduction needs to explain the main contributions of the work more clearly. The novelty of this paper is not clear.

4. Literature review techniques has to be strengthened. The authors should consider more recent research done in the field of their study.

5. The method section is not clear. Above all, it is not clear that what kinds of parameters were collected. How did you collected the parameters that were not mentioned in method section. What is the characteristic of the data. Second, it is not clear whether the image quality check was only performed on data for LS analysis? What about the other parameters?

6. There is a missing of citation for the random forest algorithm. How did you determine the key features? It is not clear about the architecture of the MLP. Also, it is not clear about the split of data for training, validation and test.

7. Result section. Correlation was presented, but where was the result of RF for parameters selection?

8. It is unclear how NC/C was generated.

9. Discussion section. what is boundary subsets?

10. Authors stated that ‘cardiac mechanics seems to be more promising as a specific marker, and trabeculae may be considered just a red flag cardiac mechanics’. However, while the cardiac mechanics parameters, ie. GLS and LVEF were the result of various phenotypes, it is not clear how the authors came to this statement. Please elaborate.

Reviewer #2: This paper uses a machine learning (ML) based model to evaluate myocardium function, and the role of echocardiographic predictors for rigid body rotation in Noncompaction cardiomyopathy (NCC). The manuscript is clear, straightforward, easy to follow. The results are sufficient and convincing. I think this study has the potential for NCC risk stratification. However, I would like to draw the author's attention to the following major concerns:

1)In this study, the authors used 50 patients for training and testing the machine learning model. Considering, the model is a deep neural network. The data set the size of 50 patients is insufficient.

2)The authors do not explain clearly its used deep neural network. As a key point, what is the architecture of the deep neural network? Does the network use a general architecture, or is it specially optimized for NCC? Do different network architectures change the results？The authors should do a more thorough literature survey. Just to name a few:

-Cardiac Functional Analysis with Cine MRI via Deep Learning Reconstruction

-Segmentation and quantification of infarction without contrast agents via spatiotemporal generative adversarial learning.

-Automated left ventricular dimension assessment using artificial intelligence developed and validated by a UK-wide collaborative

3) In this study, the authors used a single echocardiographic ultrasound system from a single center. I think this point is the most important limitation of this study. The generalization error is a big problem for machine learning (ML) based models. It would be better to discuss this point in the manuscript.

4) There are some grammar errors and typos. I suggest the authors make an solid, overall proofreading.

6. PLOS authors have the option to publish the peer review history of their article (what does this mean?). If published, this will include your full peer review and any attached files.

Reviewer #1: No

Reviewer #2: No

---

## [Author Response · Author response to Decision Letter 0]

12 Oct 2021

Dear Dr. Gao

We are thrilled by the opportunity to resubmit our revised manuscript. On behalf of all co-authors, I would like to thank you and the invited reviewers for the time and effort dedicated to reviewing our work, as well as for the revisions that we believe have strengthened this paper. 

We have made every effort to address all points and issues raised. Enclosed please see detailed response to each of the reviewers’ comments to the manuscript. We believe that the information provided in this manuscript has significant clinical value and should be communicated to the scientific community through this journal. 

 Sincerely,

---

## [Decision Letter · Decision Letter 1]

5 Nov 2021

A Machine learning framework for the evaluation of myocardial dysfunction in patients with noncompaction cardiomyopathy and rigid body rotation

PONE-D-21-10289R1

Dear Dr. Salemi,

We’re pleased to inform you that your manuscript has been judged scientifically suitable for publication and will be formally accepted for publication once it meets all outstanding technical requirements.

Kind regards,

Zhifan Gao

Academic Editor

PLOS ONE

Additional Editor Comments (optional):

Reviewers' comments:

Reviewer's Responses to Questions

**Comments to the Author**

1. If the authors have adequately addressed your comments raised in a previous round of review and you feel that this manuscript is now acceptable for publication, you may indicate that here to bypass the “Comments to the Author” section, enter your conflict of interest statement in the “Confidential to Editor” section, and submit your "Accept" recommendation.

Reviewer #1: All comments have been addressed

Reviewer #2: All comments have been addressed

2. Is the manuscript technically sound, and do the data support the conclusions?

Reviewer #1: Yes

Reviewer #2: Yes

3. Has the statistical analysis been performed appropriately and rigorously? 

Reviewer #1: Yes

Reviewer #2: Yes

4. Have the authors made all data underlying the findings in their manuscript fully available?

Reviewer #1: No

Reviewer #2: No

5. Is the manuscript presented in an intelligible fashion and written in standard English?

Reviewer #1: Yes

Reviewer #2: Yes

6. Review Comments to the Author

Reviewer #1: Authors presented an exploratory investigation of the markers that could potentially help in differentiate diagnosis for NCC patients. Significant improvements have been made. There are several minor issues

1. In Figure 1, it seems that the white line was supposed to be the difference between green line and pink line, while it is confusing in the caption that what does the “curve mean” mean?

2. It will be appreciated to see the ROC analysis for differentiating the cases with or without RBR in patients with NCC.

Reviewer #2: (No Response)

7. PLOS authors have the option to publish the peer review history of their article (what does this mean?). If published, this will include your full peer review and any attached files.

Reviewer #1: No

Reviewer #2: No

---

## [Editor Report · Acceptance letter]

11 Nov 2021

PONE-D-21-10289R1 

A machine learning framework for the evaluation of myocardial rotation in patients with noncompaction cardiomyopathy 

Dear Dr. Salemi:

I'm pleased to inform you that your manuscript has been deemed suitable for publication in PLOS ONE. Congratulations! Your manuscript is now with our production department. 

Kind regards, 

on behalf of

Dr. Zhifan Gao 

Academic Editor

PLOS ONE